# Efficacy of Laser Therapy on Paralysis and Disability in Patients with Facial Palsy: A Systematic Review of Randomized Controlled Trials [note 1]

**DOI:** 10.3390/healthcare11172419

**Published:** 2023-08-29

**Authors:** Jung-Hyun Kim, Bonhyuk Goo, Sang-Soo Nam

**Affiliations:** 1Department of Acupuncture & Moxibustion, Kyung Hee University Hospital at Gangdong, Seoul 05278, Republic of Korea; dan_mi725@naver.com (J.-H.K.); goobh99@naver.com (B.G.); 2Department of Acupuncture & Moxibustion, College of Korean Medicine, Kyung Hee University, Seoul 02447, Republic of Korea

**Keywords:** laser therapy, systematic review, facial palsy, neurological rehabilitation

## Abstract

(1) Background: Facial palsy is a common health issue which leads to sequelae and disability. This systematic review aimed to assess the efficacy of laser therapy for the treatment of facial palsy. (2) Methods: Only randomized controlled trials comparing the effectiveness of laser therapy to non-laser intervention, no intervention, or placebo were searched for. Relevant studies were searched in seven electronic databases. Studies that examined the use of laser modalities for facial palsy management, with or without acupuncture, were also included. Two authors independently read and scored the methodological quality of the selected texts, and any disagreement was resolved by discussion or by intervention from the third author. (3) Results and conclusions: With five full-text articles, a methodological quality for each included study was assessed (kappa coefficient = 0.75). The laser therapy group in the mean difference measuring FDI showed an effect size of 8.15 compared to the control group; while measuring the paralysis score, an advantage was disclosed with an effect size of 0.22 compared to the control group.

## 1. Introduction

Facial palsy is a lesion of the seventh cranial nerve that differs from central palsy in clinical terms due to the involvement of face muscles surrounding the eye [1]. Depending on the population under study, the reported yearly incidence of facial palsy ranges from 11 to 40, from 20 to 25, and from 12 to 53 incidents per 100,000 individuals [2]. Additionally, patients may receive insufficient care and suffer from contracture, hyperkinesis, or synkinesis, the latter of which can range from 1.7% to 42% in prevalence [3].

Facial creases and the nasolabial fold vanish, the forehead unfurrows, and the corner of the mouth lowers in facial palsy patients. The lower lid sags, and the eyelids do not close. Lack of lubrication and greater exposure to external elements may cause eye discomfort. Tear production reduces, but because the lids can no longer be controlled, tears might spill out freely, giving the appearance that the eye is crying excessively. On the affected side of the mouth, food and saliva might collect and leak out of the corner [4].

Patients with facial palsy typically progress from the onset of symptoms to maximal weakness within three days, and almost always within one week [5], although treatment with a 7-day course of acyclovir or valacyclovir along with prednisone is recommended to shorten the time to full recovery and increase the likelihood of complete recovery. Within three weeks of commencement, 85% of patients display at least partial recovery when untreated [6]. After this period, treatment with medication cannot be expected to reduce the sequelae or synkinesis. Therefore, complementary and alternative approaches are required.

Corticosteroids, such as prednisone, are anti-inflammatory drugs that can help to reduce swelling and inflammation of the facial nerve. Antiviral medications, such as acyclovir, are used to treat viral infections that may be causing the facial palsy. Physical therapy can help to improve facial muscle function by strengthening and stretching the muscles. It can also help to prevent contractures, which are permanent tightening of the muscles. Also, laser therapy can be considered as an alternative method. By decreasing the transmission of pain to the brain and nociceptors, laser therapy reduces pain and has anti-inflammatory effects due to the stimulation of mitochondria, stabilization of cellular membrane, and regeneration effects [7]. However, inconsistent results have been reported using current rehabilitative techniques [8]. All precedent reviews can be categorized as low-quality evidence [9]. Furthermore, a systematic review published to date has only illuminated low-level laser therapy’s impact on Bell’s palsy. Since Bell’s palsy is a temporary condition with an unknown cause, a limited approach requires attention to interpretation. Laser modalities for patients with facial palsy cannot be fully recommended due to insufficient evidence. Therefore, a study with a higher level of evidence is required to assess the effects of laser therapy in patients with facial palsy. Therefore, this systematic review aimed to examine the effects of laser therapy on patients with facial palsy.

## 2. Materials and Methods

### 2.1. Systematic Review Registration

This systematic review was registered in PROSPERO (registration number: CRD42020168753; http://www.crd.york.ac.uk/PROSPERO, accessed on 28 April 2020) and will be reported according to the guides for systematic review and meta-analysis protocol (PRISMA guidelines) [10]. Checklist compliant to guides is demonstrated upon Appendix A.

### 2.2. Inclusion/Exclusion Criteria

#### 2.2.1. Type of Participants

The current research concentrated on patients with facial palsy of either sex or age who were receiving laser treatment. Studies featuring individuals in all stages of facial palsy, including the sequelae, were included in our review. Diagnostic methods for facial palsy were not considered in the inclusion criteria.

#### 2.2.2. Type of Intervention

As long as the experimental groups did not receive any additional therapy, such as medication, the authors covered all experiments that used any kind of laser application with varied wavelengths. Studies that compared laser therapy with exercise, massage, or any type of physical therapy were included in this systematic review. Any laser acupuncture was included provided that it was a noninvasive treatment and differed from traditional acupuncture treatment that penetrates the skin with needles.

#### 2.2.3. Outcome Measure

The articles selected for this review were RCTs that used laser therapy to treat patients with facial palsy. The outcome measures considered were labial mobility, muscle stiffness, synkinesis, facial symmetry, and the psychosocial aspects of facial palsy. Studies using these measures were eligible for inclusion:(a)Facial disability indices [11]

This index consists of 10 items that assess the patients’ social and physical aspects. The social facial disability index evaluates communication, emotional alterations, and social integration. The physical and facial disability indices assess mastication, deglutition, and labial mobility. Higher scores indicate less impairment and fewer disabilities on a scale of 100.

(b)Sunnybrook facial grading system [12]

This system assesses three areas of facial asymmetry: resting asymmetry (scored from 0 to 4, with 4 indicating the most asymmetrical), symmetry of voluntary movement (scored from 0 to 5, with 5 indicating the most symmetrical), and synkinesis (scored from 0 to 3, with 3 indicating the worst). A perfect score of 100 points indicates normal facial symmetry.

(c)House and Brackmann scale [13]

Initially, this scale was categorized as a universal scale by the American Academy of Otolaryngology—Head and Neck Surgery. This was subsequently modified by House and Brackmann. This scale evaluates symmetry, synkinesis, stiffness, and global mobility of the face. Ranging from normal to total paralysis, the scale is scored from 0 to 6, with 6 indicating total paralysis.

(d)Paralysis score [14]

The paralysis score is calculated by evaluating the degree of facial paralysis according to the Japanese Facial Neurological Research Group. The degree of symmetry at rest and the state of forehead wrinkles observed on the face are divided into five stages and assigned 4 points for normal and 0 points for complete paralysis. The total score is 40 points; therefore, the greater the paralysis is, the lower the score is.

(e)Facial stiffness scale [15]

When measuring facial stiffness as a self-report measure, participants are asked to choose a number that best represented the current stiffness level on a scale of 1 to 5 (1 = no stiffness, 5 = very rigid).

#### 2.2.4. Type of Studies

Randomized controlled trials (RCT) that reported the effects of laser application on facial palsy were included in this review. The articles needed to include at least one control group that underwent treatment with modalities other than laser, or no intervention. The presence or absence of a blinding process within each study was verified in a methodological evaluation; the authors did not include only RCTs in the retrieval process.

### 2.3. Search Strategy

From 1–4 January 2023, two authors independently searched seven electronic databases (PubMed, Scopus, EBSCO CINAHL, Ovid MEDLINE, Ovid EMBASE, Cochrane Library with physical therapies, and Web of Science) to identify relevant articles published until December 2022. To identify the keywords used, facial palsy and laser therapy were searched in Medical Subject Headings, and their relevant terms were selected for the database retrieval procedure. The selected keywords were (“laser” OR “phototherapy*”) AND (“facial palsy” OR “facial paralysis” OR “facial neuropathy”). The researchers manually searched the references of the selected studies to identify other relevant studies. No language restriction was applied during the retrieval procedure. The involvement of experts in physical therapies and facial palsy was requested in the retrieval procedure.

### 2.4. Selection Process

Two authors (J.-H.K. and B.G.) independently reviewed the titles and abstracts of all the identified papers. If more than one report was found in the same study, the duplication was eliminated through discussion by the two authors. The entire texts of all potential abstracts were collected by the authors, who then independently evaluated each one in light of the inclusion/exclusion standards.

### 2.5. Strategy for Data Synthesis

By construct similarity, the primary findings of the included research were divided into the following domains:(1)Paralysis score(2)Facial disability index physical (PFDI), facial disability index social (SFDI)(3)Sunnybrook facial grading system(4)House and Brackmann scale(5)Facial stiffness scale

The published description of the development and validation of each instrument were retrieved as part of this process. Consensus among the authors was used to categorize the domains, and any outcomes deemed to fall outside of these construct categories were disregarded.

### 2.6. Methodological Quality Assessment

Two authors (J.-H.K. and B.G.) independently read the selected texts in full, appraised them critically, and scored their methodological quality using the PEDro quality assessment tool [16]. Any disagreement between the authors was discussed, and the final scores were obtained. If the disagreement was not resolved, the third author (S.-S.N.) intervened.

### 2.7. Data Collection

The following data were separately gleaned from the trials by two reviewers: each study’s design and blinding, sample size, disease stage, accompanying interventions for both the experimental and control groups, number of treatment sessions, duration and frequency of treatment, technical characteristics of the laser, outcome measures, assessment period, and final results. These were all extracted by two authors (J.-H.K. and B.G.) who independently read the full texts of the included articles. Cochrane software Review Manager 5.0.24 was selected to establish forest plots comparing outcomes between the experimental and control groups.

## 3. Results

We identified 755 articles in the selected databases based on the keywords mentioned above. After preliminary retrieval, 176 potentially relevant abstracts were selected for a full text review. Finally, five articles were identified and critically appraised by two independent authors. The inclusion requirements were not met by a total of 750 articles; thus, they were excluded. The most frequent justifications for excluding studies were that they did not include the type of patients we focused on, there was no control group, and two or more combined interventions were used in the experimental group.

If the full text was not accessible online, a manual search was conducted to access the actual publication. No studies were performed following this manual search. A flowchart of the retrieval procedure and selection criteria is shown in Figure 1.

### 3.1. Methodological Quality of Selected Articles

The PEDro scale scores of the included articles ranged from 4 to 7. Points are awarded when a criterion is clearly satisfied. Details of the quality assessment compliant with the PEDro scale are presented in Table 1.

### 3.2. Characteristics of the Selected Articles

#### 3.2.1. Laser Properties of the Selected Studies

All of the recognized facilitators in the chosen publications used the pencil-like technique on the facial nerve roots, and the majority of their therapy sessions used wavelengths of 830 nm. The number of emitters, kind of emitter, beam delivery system, center wavelength, spectral bandwidth, energy emission per pulse, polarization, average radiant power, and beam divergence were not all given in the considered studies.

#### 3.2.2. Summarized Results

In Murakami’s study [17], 26, 11, and 15 patients underwent stellate ganglion blocking (SGB), low-level infrared (830 nm) diode laser therapy, and a combination of treatments, respectively. To compare the effects of the three domains, the data were evaluated. Patients who received SGB alone or SGB combined with LLLT or LLLT alone demonstrated a comparable overall recovery from paralysis. The paralysis scores initially improved only a little bit more in the LLLT group. In the LLLT group, grave side effects were not seen.

Yamada [18] compared the cases of seven patients who received laser treatment, ten who were prescribed corticosteroids, and seven who had both. Patients who received laser treatment generally showed very similar recovery to those who received corticosteroid treatment; the best recovery was achieved in the shortest amount of time for individuals who underwent combination treatment. There were no side effects of LLLT that were clinically noteworthy.

In Alayat’s study [19], participants were randomly assigned to the HILT, LLLT, and exercise groups. Using the House and Brackmann scale and the Face Disorder Scale, the degree of facial healing was assessed. Three and six weeks after therapy, assessments were carried out. This study demonstrated that HILT and LLLT greatly accelerated Bell’s palsy patients’ recovery. In addition, HILT was more effective than LLLT or massage. Between LLLT and HILT, the latter demonstrated a slightly greater improvement than the former. Using the facial disorder index (FDI) before treatment, as well as three and eight weeks after treatment, the palsy improvement rate was assessed. The FDI scores in the exercise group did not differ significantly between the start of treatment and week three (*p* > 0.05), but week six (*p* > 0.001) saw a considerable improvement. Significant improvements in FDI scores at baseline in the laser group were observed at weeks three and six (*p* < 0.001). The improvement in FDI scores was significantly greater in the laser group than in the exercise group (*p* < 0.05) at 3 and 6 weeks. According to our research, treatment combining LLLT and exercise therapy is linked to significantly better FDI results than exercise therapy alone.

Ordahan’s study [20] aimed to investigate the effectiveness of LLLT during the initial recovery period in patients with peripheral facial palsy. Patients in the experimental group underwent LLLT and facial exercise therapy, whereas those in the control group underwent facial exercise therapy alone. A gallium-aluminum-arsenide (GaAlAs) diode laser was used to provide the laser therapy at an 830 nm wavelength, 100 mW power, and 1 kHz frequency. An average energy density of 10 J/cm^2^ was administered thrice a week to eight affected points on the face for six weeks. The FDI was used to assess the facial improvement rate before therapy, as well as three and six weeks following therapy. Although there was no significant change in the FDI scores between the baseline and week three (*p* > 0.05) in the exercise group, a substantial improvement was seen at week six (*p* ≤ 0.001). At weeks three and six, there was a noticeable increase in FDI scores in the laser therapy group compared to baseline (*p* ≤ 0.001). At weeks three and six, improvements in FDI scores were noticeably higher in the laser group than in the exercise group (*p* ≤ 0.05).

In Ton [21]’s study, 17 participants were divided into two groups: those receiving laser acupuncture therapy (LAT) and those receiving phony laser acupuncture therapy (sham LAT). Three times per week for six weeks (18 treatments) was the schedule for the LAT group, whereas the sham LAT group performed the identical process using fake laser equipment. The change in the social domain of FDI from baseline to week six was the primary outcome. The secondary outcomes were the changes in the House-Brackmann (HB) grade, Sunnybrook rating system (SB), and 3- and 6-week facial stiffness scale scores. Between the baseline and three weeks, the experimental group’s HB grades significantly differed (*p* = 0.0438), and between the baseline and six weeks, the SB and face stiffness scores exhibited minimal statistical significance (*p* = 0.0598 and *p* = 0.0980, respectively). In week six, there was no discernible difference between the reference point and the FDI score.

All results of the included studies are summarized in Table 2.

### 3.3. Data Synthesis

Although five outcome measures were used in the included studies, only the paralysis score and FDI were synthesized by comparing the mean and standard deviation. The laser therapy group in the mean difference measuring FDI showed an effect size of 8.15 compared to the control group; while measuring the paralysis score, an advantage was disclosed with an effect size of 0.22 compared to the control group. The planned meta-analysis was not proceeded because the number of eligible studies was too small to conduct a valid meta-analysis.

## 4. Discussion

Although facial palsy leads to visible disharmony and a poor quality of life, there is not enough information to determine which treatment is most likely to be safe and effective. Healthcare decision-makers may be confused because the majority of previously published articles are not RCTs, and the quality of control group setting is poor.

To overcome the limitations of existing articles, this review expanded the category of laser treatment from LLLT and checked the outcome measures that were not included in the published paper by manually contacting the corresponding authors. The studies included in this review were heterogeneous in the design, type and content of interventions, measurement of outcomes and effects. 

In this study, the PEDro scale was used to determine the risk of methodological bias [22]. Assignment concealment and intention-to-treat analyses were factors that could increase the risk of bias [23]. To prevent baseline characteristic discrepancies caused by biased participant selection or assignment, allocation concealment is required [24]. Intention-to-treat analysis is needed to prevent bias due to patient loss, which can affect the criteria equivalence [25].

According to recent research, patients with facial palsy can benefit from using 830 nm LLLT with 100 mW output and 120 s duration on eight sites of the afflicted site’s facial branches for six weeks [26]. However, caution should be exercised when generalizing the results of this study. This is because the number of RCTs was small, and the difference between the groups was not large in the two RCTs. The forms of laser treatment other than LLLT must be verified for their effectiveness.

Laser therapy used in this study can be divided into LLLT, HILT, and laser acupuncture. Pulsed Nd:YAG lasers created by HIRO 3 devices (ASA, Arcugnano, Vicenza, Italy) were used to treat the HILT group. The device emitted a pulse of 1064 nm and exhibited a higher peak power than LLLT (3 kW). During the HILT sessions, high levels of energy density (810–1780 mJ/cm^2^), brief durations (120–150 s), low frequencies (10–40 Hz), and duty cycles of roughly 0.1% were displayed. The afflicted domain of the facial nerve surface received HILT treatment in a perpendicular direction. An energy density of 10 J/cm^3^ was used. An entire 80 J of energy was given to the patients during a single session. HILT was applied in 18 sessions for 6 consecutive weeks (three times/week).

In the LLLT sessions, an infrared probe with a wavelength of 830 nm and an output of 100 mW was used. A GaAlAs laser with a lower energy density (≤10 J/cm^2^), 1 kHz frequency, and 80% duty cycle was used. The damaged area’s roots of the facial nerve surface were exposed to laser radiation while the surgery was performed at eight points for 2 min and 5 s per point. LLLT was applied for 18 sessions over 6 consecutive weeks (three times/week). Calibration of the laser equipment was performed by the manufacturer using a heat output meter.

LAT is described as the photonic stimulation of acupuncture sites using low-intensity, nonthermal laser irradiation, as opposed to LLLT, which uses the photonic stimulation of specific locations to produce therapeutic effects in the body [27]. LAT does not physically penetrate the skin, which is one way that it differs from acupuncture needles. Therefore, LAT can be categorized as a type of laser therapy. Previous studies have reported neural modulatory effects of LAT [28,29,30]. Although a few side effects, including brief drowsiness, headaches, and fatigue, have been associated with LAT [31], no adverse effects were reported in the included studies. Since LAT makes it possible to stimulate areas that are challenging or uncomfortable to treat with a needle [32], it can also be considered as an alternative treatment for facial palsy.

Our systematic review has some limitations. First, the study only comprised clinical trials with controls, so there is a risk of biased generalization when interpreting the findings. Second, the dependability of the included publications’ results in the analysis process was weak because all of them utilized a non-parametric test. Because there were not enough samples in the listed studies, a non-parametric test was applied. Also, the expected meta-analysis was abandoned since there were not enough suitable trials for a reliable result. Additionally, the reliability and validity of the measured outcome measure could not be fully demonstrated due to insufficient data. Finally, all papers included in this study observed results within six weeks, making it difficult to verify the effects of laser treatment for longer periods, for example, for chronic facial palsy, sequelae, or the mitigation of facial synkinesis.

Blinding, allocation concealment, randomization, and statistical analysis should all be carried out as part of the study design in order to avoid these restrictions and employ a sham laser. The effectiveness of laser therapy for treating persistent facial palsy or the after-effects of facial palsy requires further research.

## 5. Conclusions

Several studies have cautiously suggested that laser therapy can effectively improve the symptoms and discomfort in patients with facial palsy. Laser therapy may be effective in improving facial nerve function and reducing the severity of symptoms in patients with facial palsy. The quality of the evidence is limited, and more research is needed to confirm the effectiveness of laser therapy for facial palsy. The optimal type and parameters of laser therapy for facial palsy are not yet known. Well-designed RCTs with larger sample sizes are required in the future.

## Figures and Tables

**Figure 1 healthcare-11-02419-f001:**
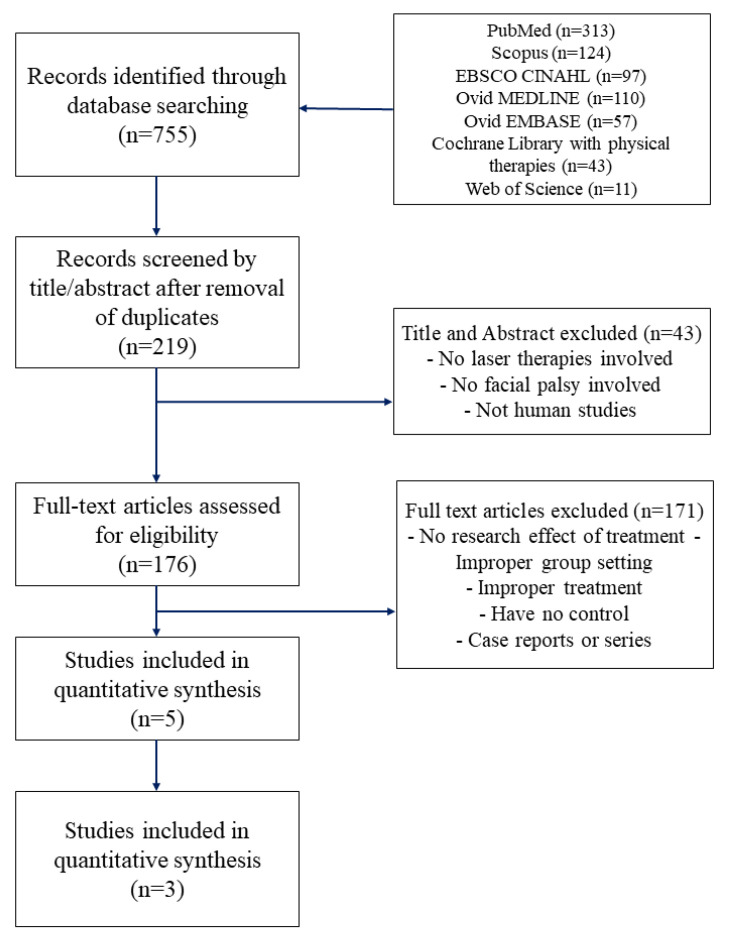
Flowchart of the retrieval process.

**Table 1 healthcare-11-02419-t001:** Quality retrieval of the included studies based on the PEDro scale.

Sections	Study ID				
	Murakami et al., 1993 [17]	Yamada et al., 1995 [18]	Alayat et al., 2013 [19]	Ordahan 2016 [20]	Ton et al., 2021 [21]
Eligibility criteria specification	Yes	Yes	Yes	Yes	Yes
Proper random allocation	No	Yes	Yes	No	Yes
Allocation concealment	No	No	Yes	Yes	Yes
Group similarity at baseline	Yes	No	Yes	No	Yes
Subjects blinding	No	No	Yes	Yes	No
Therapists blinding	No	No	No	No	No
Assessor blinding	No	No	No	No	No
Outcome measure obtained from 85% of subjects initially allocated to groups	Yes	Yes	Yes	Yes	Yes
Use of intention-to-treat analysis	No	No	No	No	No
Reporting between group statistical comparisons	Yes	Yes	Yes	Yes	Yes
Reporting point measures and measure of variability	Yes	Yes	Yes	Yes	Yes
Total	4	4	7	5	6

**Table 2 healthcare-11-02419-t002:** Research characteristics of the included studies.

Study	Participants	Interventions	Outcome Measures	Results/Conclusions
Murakami et al., 1993 [17]	Eleven patients received LLLT, 15 received SGB therapy, and 26 received combined therapy.	A GaAIAs ^1^ semiconductor model (MLD-1003) with 830 nm wavelength and 150 mW was used. The probe was irradiated to a skin surface.	The facial paralysis scale was used. The total score is 40 points. The greater the paralysis is, the lower the score is. Paralysis scores were measured at weeks 0, 2, and 6.	The laser therapy group showed that the average change in the paralysis score was 10.1 before treatment, 14.6 during the second week of treatment, 33.3 at the end of treatment, and 34.4 during laser treatments.
Yamada et al., 1995 [18]	Twenty-four patients with facial palsy were selected for this study. Seven patients received LLLT only. Seven patients received LLLT and CS. Ten patients received CS only.	GaAIAs and the MLD-1003 system (Mochida, Tokyo, Japan) were used for the LLLT.	Ten-section scale devised by the Facial Nerve Research Group. This scale was used to assess the degree of facial palsy at the time of initial consultation, two weeks after treatment had commenced, and again when treatment was terminated.	After two weeks, when the corticosteroid administration was complete, the score in the LLLT group had improved (10.0 to 14.6). The score in the combination group had improved from 9.3 to 23.4, while the score in the CS group had improved from 10.4 to 20.5.
Alayat et al., 2013 [19]	3 groups of 17 patients each: – HILT group: HILT, facial massage, and facial expression exercise – LLLT group: LLLT, facial massage, and facial expression exercise – Exercise group: facial massage and facial expression exercise with sham laser therapy	All patients were treated with facial massage and exercise, but the HILT and LLLT groups received different forms of laser treatment, respectively. The afflicted portion of the face received eight laser pointers, three times per week for six weeks.	To assess the grade of recovery, the FDI and HB scale were used. The scores of both FDI and HBS were taken before and three and six weeks after treatment.	The HILT and LLLT groups showed improvement in HBS and FDI scores after 3 and 6 weeks of treatment, with the greatest improvement at 6 weeks. LLLT group showed the largest improvement in FDI score, followed by HILT group. The exercise group showed the least improvement. The HILT group showed the best efficacy for HBS.
Ordahan 2016 [20]	Forty-six patients (average age 41 ± 9.7 years, 40 females, six males)	A GaAIAs with LLLT was utilized.	FDI was observed at weeks 0, 3, and 6.	The FDI scores were similar in the comparison and control groups before treatment. The exercise group showed no significant improvement in FDI scores at week 3, but significant improvement was observed at week 6. The laser treatment group showed significant improvement in FDI scores at weeks 3 and 6, and the degree of improvement was superior to that of the exercise group.
Ton et al., 2021 [21]	Six patients underwent LAT, while eight patients underwent sham LAT during treatment sessions	For six weeks, three times a week, laser therapy sessions were conducted for a total of 18 sessions. Typical laser units consist of visible but harmless infrared beams.	FDI, HB, SB, and facial stiffness measurements were performed on sessions one, nine, and 18 and compared with the reference score.	There were significant changes in the HB scores in the laser acupuncture groups at week three of treatment, unlike in the sham laser acupuncture group.

^1^ GaAIAs refers to gallium-aluminum-arsenide diode laser system.

## Data Availability

Data available on request from the authors.

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
