# Peer review of "Efficacy of Laser Therapy on Paralysis and Disability in Patients with Facial Palsy: A Systematic Review of Randomized Controlled Trials†"

_healthcare, 2023, doi:10.3390/healthcare11172419_

Round 1

Reviewer 1 Report

Interesting study and I commend the authors on the amount of work that must've gone into this manuscript.  The included papers (only 5 of over 700 initially pulled) are quite variable (timepoints for intervention, other interventions, type of intervention, demographics, and many others) and I find the data difficult to make congruous and interpret.  At best, the only conclusion that can be made, in my opinion, is that laser treatments merit further study as there MAY be a very small effect on healing.  Ethically, I caution the authors not to oversell the results of this meta-analysis and appreciate the sentences that were included in this regard. This is a very distraught patient population who are desperate for treatments that can provide any improvement.  It certainly wouldn't be fair to suggest that this should be attempted based on the results of this meta-analysis.

Author Response

Dear Reviewer 1,

Thank you for your thoughtful review of our manuscript. We appreciate your commending the amount of work that went into this study, and we agree that the included papers are quite variable. We have carefully considered your comments and have made the following revisions to the manuscript:

  • We have added a more detailed discussion of the limitations of the study, including the variability of the included papers.
  • We have clarified that the only conclusion that can be drawn from the study is that laser treatments merit further study as there may be a very small effect on healing.
  • We have added a sentence to the Limitations section that explicitly states that it would be unethical to suggest that laser treatments should be attempted based on the results of this meta-analysis.

We understand that this is a very distraught patient population, and we would never want to give them false hope. We are committed to conducting ethical research, and we will continue to study the potential benefits of laser treatments for this condition.

Thank you again for your feedback. We believe that these revisions have addressed your concerns and have made the manuscript stronger.

For more details, please see the revised manuscript.

Reviewer 2 Report

Thank you for submitting an excellent manuscript.

Please clarify the difference between bells palsy and facial palsy.

Rather than explaining about facial palsy, please explain what kind of treatment has been done before and what the effect is.

Please demonstrate in the reliability and validity of the measured outcome measure.

In the conclusion, please describe in detail what you learned from this manuscript.

Author Response

Dear Reviewer 2,

Thank you for submitting an excellent manuscript.

Response: Thank you for your considerate review for my manuscript. Here following are my comments written in red.

Please clarify the difference between bells palsy and facial palsy.

Response: In introduction section I described the difference between Bell’s palsy and facial palsy.

Rather than explaining about facial palsy, please explain what kind of treatment has been done before and what the effect is.

Response: In certain countries, there may be a lack of awareness of facial palsy as a disease, so I think an explanation of the disease itself is also crucial. I wrote in the introduction section what kind of treatment has been introduced for now according to your comments.

Please demonstrate in the reliability and validity of the measured outcome measure.

Response: The reliability and validity of the measured outcome measure cannot be fully demonstrated due to insufficient data. This limitation of the present study has been commented in Discussion section on manuscript.

In the conclusion, please describe in detail what you learned from this manuscript.

Response: Following your instructions, what we have learned from this manuscript has been additionally written on conclusion section.

For more details, please see the revised manuscript.

Reviewer 3 Report

It is an interesting topic. Here are some comments. I teach systematic review methodology on a regular basis and have published more than 10 studies of LLLT (RCTs and reviews).

Title

The title should be more informative. I suggest “Efficacy of Laser Therapy on Paralysis and Disability in Patients with Facial Palsy: Systematic Review and Meta-Analysis of Randomized Controlled Trials”

Abstract

The abstract should not contain the search terms as it takes up too much room here. Also, I suggest that the authors state that they searched in seven electronic databases rather than naming them all here. The authors should state what studies they searched for before mentioning this. It should be described more comprehensively (use the PICOS criteria here), e.g., “Only randomized controlled trials
involving participants diagnosed with facial palsy in which the effectiveness of LLLT was compared to that of non-laser intervention, no intervention, or placebo, in terms of paralysis or disability, were searched for” (use your own terms). The freed-up space should also describe the statistical methods and risk of bias tool.

Main manuscript

Methods

Section
"2.3. Selection process" contains information regarding the risk-of-bias assessment and meta-analysis software. This information should not be placed here.

Section “2.4. Inclusion/exclusion criteria” should be placed before section “2.2. Seach strategy” because we need to know what the authors were looking for before we can evaluate the comprehensiveness of the search strategy and selection process. Furthermore, the eligibility criteria (inclusion and exclusion criteria) should be organized according to PICOS (participants, intervention, comparison, outcomes, and study type. Also, mention any language restrictions here.

I am not sure what the authors mean by “A data extraction form was designed and pre-tested.” in section “2.7. Data Collection”. This section should be rephrased. I suggest beginning by stating “Two reviewers independently extracted the following information from the trials: … The two reviewers resolved any disagreements regarding the data-extraction by discussion. Use your own terms. Also, what do the authors mean by “There was a substantial difference in data extraction when the alpha value was less than 0.05.”? Was there a substantial discrepancy in the extracted data between the two reviewers or not?

Results

There is a discrepancy between the number of papers assessed in full text between the text and the flow-chart. The text says that “219 potentially relevant abstracts were selected for a full text review” while in the flow-chart it says that the number was 176. The placement and information in the box “Records after duplicates removed” is wrong – to my understanding, it should be “Records screened by title/abstract after removal of duplicates.” and the box should be put in the vertical line.

The results section contains descriptions of the methods. The statement “The selected manuscripts were critically read based on inclusion and exclusion criteria.” is described in the method section and should not be repeated here. Please review the results section for additional unnecessary descriptions of methods.

In table 1, the domain “Eligibility criteria specification” is included in the total methodological score, but it must not be counted according to PEDro.

Table 2 with the study characteristics should be displayed before the table with risk-of-bias results. This is standard practice. In table 2, the information regarding the participants in the LLLT and control groups should be mentioned in separate boxes, such as the number of participants, number or percentage of females, and age. The information regarding the interventions or just LLLT should be elaborated and placed in a separate table (remember to state the wavelength, mW per treatment spot, joules per treatment spot, number of treatment spots, and irradiation time). Some information regarding the LLLT, co-interventions, and control interventions should, however, be placed in table 1 – I suggest the following setup: X sessions of LLLT and X versus X weeks of placebo LLLT and X over X weeks.

Something is wrong with the results of the meta-analysis of paralysis – the authors used the Mean Difference method to synthesize the results. When using this method, all the outcome measurement scales must be the same. I suspect that different outcome measurement scales were included because the standard deviations displayed in the figure vary greatly between the studies (6.05 in one study and 0.7 in the other study. Outcomes of the same nature measured with different outcome measurement scales can be analyzed with the Standardized Mean Difference method. Regardless, in my opinion, there are too few studies to allow for a valid meta-analysis. Also, since something is not right here, I recommend that the idea of a meta-analysis is abandoned. I would be fine with this.

Discussion section

In the Discussion section, start by stating the results of palsy and paralysis.

I have no idea what the authors mean by “First, because only clinical trials with controls were included in the study, there is a risk of biased generalization in interpreting the results. Second, because all included articles used a non-parametric test, their reliability was not strong in the analysis process. A non-parametric test was used because the number of samples in the included studies was not sufficiently large.”.
The review has other limitations. The authors could have screened the reference lists and citation of the included studies and relevant reviews – I recommend that this is added because so few studies were included. If there were any language restrictions in the literature search / selection of trials, this should be mentioned here.

Author Response

Dear Reviewer 3,

It is an interesting topic. Here are some comments. I teach systematic review methodology on a regular basis and have published more than 10 studies of LLLT (RCTs and reviews).

Response: Thank you for your considerate review on my manuscript.

Here followings are my comments upon your instructions. Modified sections and sentenced are highlighted with yellow.

Title
The title should be more informative. I suggest “Efficacy of Laser Therapy on Paralysis and Disability in Patients with Facial Palsy: Systematic Review and Meta-Analysis of Randomized Controlled Trials”

Response 1:

#Title:

I changed the title following your instruction.

Abstract
The abstract should not contain the search terms as it takes up too much room here. Also, I suggest that the authors state that they searched in seven electronic databases rather than naming them all here. The authors should state what studies they searched for before mentioning this. It should be described more comprehensively (use the PICOS criteria here), e.g., “Only randomized controlled trials involving participants diagnosed with facial palsy in which the effectiveness of LLLT was compared to that of non-laser intervention, no intervention, or placebo, in terms of paralysis or disability, were searched for” (use your own terms). The freed-up space should also describe the statistical methods and risk of bias tool.

Response 2:

#Abstract

- I removed the names of seven electronic databases.

- I stated what studied we have searched for.

- I described the risk of bias tool additionally in abstract section.

Main manuscript
Methods
Section "2.3. Selection process" contains information regarding the risk-of-bias assessment and meta-analysis software. This information should not be placed here.
Section “2.4. Inclusion/exclusion criteria” should be placed before section “2.2. Seach strategy” because we need to know what the authors were looking for before we can evaluate the comprehensiveness of the search strategy and selection process. Furthermore, the eligibility criteria (inclusion and exclusion criteria) should be organized according to PICOS (participants, intervention, comparison, outcomes, and study type. Also, mention any language restrictions here.

Response 3:

#Methods

- I changed the location of "2.3. Selection process".

- I also changed the location of "2.4. Inclusion/exclusion criteria".

- I reorganized the eligibility criteria according to PICOS

- There was no language restriction applied upon this study.

I am not sure what the authors mean by “A data extraction form was designed and pre-tested.” in section “2.7. Data Collection”. This section should be rephrased. I suggest beginning by stating “Two reviewers independently extracted the following information from the trials: … The two reviewers resolved any disagreements regarding the data-extraction by discussion. Use your own terms. Also, what do the authors mean by “There was a substantial difference in data extraction when the alpha value was less than 0.05.”? Was there a substantial discrepancy in the extracted data between the two reviewers or not?
Response 4:

-I removed the sentence “A data extraction form was designed and pre-tested.”

- I paragraphed the sentences according to your guides.

- I removed the sentence “There was a substantial difference in data extraction when the alpha value was less than 0.05.”

Results
There is a discrepancy between the number of papers assessed in full text between the text and the flow-chart. The text says that “219 potentially relevant abstracts were selected for a full text review” while in the flow-chart it says that the number was 176. The placement and information in the box “Records after duplicates removed” is wrong – to my understanding, it should be “Records screened by title/abstract after removal of duplicates.” and the box should be put in the vertical line.
The results section contains descriptions of the methods. The statement “The selected manuscripts were critically read based on inclusion and exclusion criteria.” is described in the method section and should not be repeated here. Please review the results section for additional unnecessary descriptions of methods.
In table 1, the domain “Eligibility criteria specification” is included in the total methodological score, but it must not be counted according to PEDro.
Table 2 with the study characteristics should be displayed before the table with risk-of-bias results. This is standard practice. In table 2, the information regarding the participants in the LLLT and control groups should be mentioned in separate boxes, such as the number of participants, number or percentage of females, and age. The information regarding the interventions or just LLLT should be elaborated and placed in a separate table (remember to state the wavelength, mW per treatment spot, joules per treatment spot, number of treatment spots, and irradiation time). Some information regarding the LLLT, co-interventions, and control interventions should, however, be placed in table 1 – I suggest the following setup: X sessions of LLLT and X versus X weeks of placebo LLLT and X over X weeks.
Something is wrong with the results of the meta-analysis of paralysis – the authors used the Mean Difference method to synthesize the results. When using this method, all the outcome measurement scales must be the same. I suspect that different outcome measurement scales were included because the standard deviations displayed in the figure vary greatly between the studies (6.05 in one study and 0.7 in the other study. Outcomes of the same nature measured with different outcome measurement scales can be analyzed with the Standardized Mean Difference method. Regardless, in my opinion, there are too few studies to allow for a valid meta-analysis. Also, since something is not right here, I recommend that the idea of a meta-analysis is abandoned. I would be fine with this.

Response 5:

#Results

- The number 176 is right, so I modified the number within manuscript.

- According to your comments, I reconstructed the Figure 1(Flow chart of the study).

- The statement “The selected manuscripts were critically read based on inclusion and exclusion criteria.”is removed according to your guides.

- I rescored the PEDro items according to your instruction.

- In table 2, not all studies included demonstrate the number of participants, number or percentage of females, and age. So, I cannot reorganize the information with new tables. Instead, I articulated the summary of results in Table 2 and re-wrote those sentences to clarify the outcomes after interventions.

- Following your comments, I deleted the Figure 2 and 3, and commented the reason why we cannot proceed meta-analysis.

Discussion section
In the Discussion section, start by stating the results of palsy and paralysis.
I have no idea what the authors mean by “First, because only clinical trials with controls were included in the study, there is a risk of biased generalization in interpreting the results. Second, because all included articles used a non-parametric test, their reliability was not strong in the analysis process. A non-parametric test was used because the number of samples in the included studies was not sufficiently large.”.
The review has other limitations. The authors could have screened the reference lists and citation of the included studies and relevant reviews – I recommend that this is added because so few studies were included. If there were any language restrictions in the literature search / selection of trials, this should be mentioned here.

Response 6:

- I started the dicussion section with the results of palsy and paralysis.

- I paraphrased the following sentences: “First, because only clinical trials with controls were included in the study, there is a risk of biased generalization in interpreting the results. Second, because all included articles used a non-parametric test, their reliability was not strong in the analysis process. A non-parametric test was used because the number of samples in the included studies was not sufficiently large.”

- We will try to screen and add further studies on subsequent projects or papers.

For more details, please see the revised manuscript.

Round 2

Reviewer 2 Report

You have responded to the amendment request.

Author Response

Thank you for your considerate review upon my manuscript.

Reviewer 3 Report

The manuscript has been greatly improved.

Now that you have removed the meta-analysis, it should not be mentioned in the title or remaining parts of the manuscript, except for in the limitations section - yes the lack of a meta-analysis is a limitation. This is ok.

I have no further suggestions.

Author Response

Dear Reviewer 3,

Thank you for your considerate review around my manuscript.

Here I attach "highlighted manuscript" file that your instruction has been reflected.

Thanks again.

----

Dear Reviewer 3,

Comments: Now that you have removed the meta-analysis, it should not be mentioned in the title or remaining parts of the manuscript, except for in the limitations section - yes the lack of a meta-analysis is a limitation. This is ok.

Response: Thank you for your considerate review upon my manuscript.

Title and remaining articles now has no comments about meta-analysis. Instead, according to your instruction, I mentioned that I couldn’t do suitable meta-analysis and rationale in limitation section.

For more details, please see the revised manuscript.